# Strengthening the Immune System and Reducing Inflammation and Oxidative Stress through Diet and Nutrition: Considerations during the COVID-19 Crisis

**DOI:** 10.3390/nu12061562

**Published:** 2020-05-27

**Authors:** Mohammed Iddir, Alex Brito, Giulia Dingeo, Sofia Sosa Fernandez Del Campo, Hanen Samouda, Michael R. La Frano, Torsten Bohn

**Affiliations:** 1Nutrition and Health Research Group, Population Health Department, Luxembourg Institute of Health, 1A-B, rue Thomas Edison, L-1445 Strassen, Luxembourg; Mohammed.Iddir@lih.lu (M.I.); or abrito@labworks.ru (A.B.); Sofia.SosaFernandez@lih.lu (S.S.F.D.C.); Hanene.Samouda@lih.lu (H.S.); 2Laboratory of Pharmacokinetics and Metabolomic Analysis, Institute of Translational Medicine and Biotechnology. I.M. Sechenov First Moscow Medical University, Trubetskay Str. 8, 119991 Moscow, Russia; 3Independent Researcher, Val de Marne, 94999 Paris, France; giulia.dingeo@gmail.com; 4Department of Food Science and Nutrition, California Polytechnic State University, 1 Grand Avenue, San Luis Obispo, CA 93407, USA; mlafrano@calpoly.edu; 5Center for Health Research, California Polytechnic State University, 1 Grand Avenue, San Luis Obispo, CA 93407, USA

**Keywords:** macronutrients, trace elements, nutrient, protein intake, innate immune system, cytokines, reactive oxygen species, transcription factors, nuclear factors, infection, coronavirus

## Abstract

The coronavirus-disease 2019 (COVID-19) was announced as a global pandemic by the World Health Organization. Challenges arise concerning how to optimally support the immune system in the general population, especially under self-confinement. An optimal immune response depends on an adequate diet and nutrition in order to keep infection at bay. For example, sufficient protein intake is crucial for optimal antibody production. Low micronutrient status, such as of vitamin A or zinc, has been associated with increased infection risk. Frequently, poor nutrient status is associated with inflammation and oxidative stress, which in turn can impact the immune system. Dietary constituents with especially high anti-inflammatory and antioxidant capacity include vitamin C, vitamin E, and phytochemicals such as carotenoids and polyphenols. Several of these can interact with transcription factors such as NF-kB and Nrf-2, related to anti-inflammatory and antioxidant effects, respectively. Vitamin D in particular may perturb viral cellular infection via interacting with cell entry receptors (angiotensin converting enzyme 2), ACE2. Dietary fiber, fermented by the gut microbiota into short-chain fatty acids, has also been shown to produce anti-inflammatory effects. In this review, we highlight the importance of an optimal status of relevant nutrients to effectively reduce inflammation and oxidative stress, thereby strengthening the immune system during the COVID-19 crisis.

## 1. Introduction

A cluster of pneumonia cases caused by a previously unknown virus were noted in December 2019, in Wuhan City, China [1]. This virus is now well-known as the severe acute respiratory syndrome coronavirus-2 (SARS-CoV-2), resulting in the development of the coronavirus disease 2019 (COVID-19) [2]. The disease has spread worldwide and has been classified by the World Health Organization (WHO) as a global pandemic [2,3]. SARS-CoV-2 manifestation might be asymptomatic or moderate to severe with coughing, fever, and shortness of breath [3]. In more severe cases, complications can include acute respiratory distress syndrome, acute cardiac complications, multiple organ dysfunction syndrome, septic shock, and death [4,5,6,7]. These complications are believed to be related to what has been described as the cytokine storm, in which viral replication triggers an abnormally strong release of cytokines and other immune-related stimuli, resulting in hyper-inflammation [8].

The outbreak of this emerging infectious disease has been evolving rapidly. Strict national policies to control the disease have been implemented, including policies to practice social distancing and encouraging or even forcing people to stay at home. Especially during this self-confinement, often perceived as stressful, individuals are frequently at a loss regarding optimal dietary patterns and adequate nutrient status in order to stay healthy. In order to prevent infection, a healthy functional immune system is paramount, and an important foundation for an optimal immune response is an adequate and balanced diet [9,10,11,12,13,14].

It is well acknowledged that a low protein status can increase the risk of infection, related to, for example, low antibody production [15]. An optimal nutritional status is also fundamental to modulate inflammatory and oxidative stress processes, which are all interrelated with the immune system [16]. The important notion of the relationship between dietary constituents, nutrition, inflammation, and oxidative stress is well-regarded, and has been emphasized, for example, in the development of the anti-inflammatory dietary index [17]. Dietary and nutritional constituents known to exert anti-inflammatory and antioxidant properties include omega-3 fatty acids [18], vitamin A [19], vitamin C [20], as well as a variety of phytochemicals, such as polyphenols [21] and carotenoids [22] that are widely present in plant-based foods. Also, the dietary fiber present in plant-based food items has been associated with various health benefits including anti-inflammatory properties [11], through fermentation by the gut microbiota and consequent formation of metabolic compounds, especially short chain fatty acids (SCFA). Such anti-inflammatory active compounds could be important in the overall homeostasis of inflammation and oxidative stress, both before and/or during acute infection. In fact, dietary fiber [23] and a variety of phytochemicals such as polyphenols [24] have been proposed to influence the gut microbiota, having prebiotic effects such as fostering the growth of bacteria that are associated with health benefits, such as of *Bifidobacterium* spp., and reducing potential pathogenic ones such as *Clostridium* spp. Such aspects are of interest as gastro-intestinal complications such as diarrhea have been reported following SARS-CoV-2 infection [25]. In addition to the interrelation of nutrients and infections via inflammation and oxidative stress, additional pathways may play a role. While the vitamin A metabolite retinoic acid interacts with the transcription factor RAR (retinoic acid receptor), which may play a role in immunity, vitamin D has been proposed to interact with its own transcription factors (vitamin D receptor) or the cellular receptor important for viral entry, i.e., ACE2 (angiotensin converting enzyme 2), inhibiting virus particles from entering the cell [26].

In this review, we highlight the importance of an optimal nutrient status to strengthen the immune system during the COVID-19 crisis, focusing on the most relevant constituents that reduce inflammation and oxidative stress.

## 2. The Immune System, COVID-19, Inflammation, and Oxidative Stress

The immune response is strongly modulated by oxidative stress and inflammatory processes [27]. The non-specific or innate natural defense mechanism is derived from cells of the myelocytic line, and provides an immediate response [28]. If pathogens (i.e., viruses, bacteria) invade the body, the innate response together with the specific or adaptive defense mechanism derived from cells of the lymphocyte line, adapt their response by secreting proteins directed towards intra- and extra-cellular pathogens, including several cytokines and chemokines released by macrophages, triggering inflammation to enhance the response [29]. Inflammation and oxidative stress also contribute to the normal functioning of the human body. In particular, oxidative stress plays an essential role in mitochondrial processes [30,31,32].

Oxidative stress is predominated by an imbalance of reactive oxygen (ROS) and reactive nitrogen species (RNS), including singlet oxygen, lipid peroxides, nitric oxide, versus a decreased function of antioxidant activity or compounds, such as endogenous antioxidants (e.g., albumin, urea, reduced glutathione), exogenous antioxidants (i.e., vitamin E, vitamin C, polyphenols, carotenoids), and endogenous enzymes (superoxide dismutase (SOD), catalase (CAT), glutathione peroxidase (GPx)), among others [33]. The role of oxidative stress during infection is not fully elucidated, but free radicals have been shown to protect against invading microorganisms [34]. Chronically elevated oxidative stress occurs in long-lasting viral infections, for example, with the Epstein–Barr virus (EBV) and the human immunodeficiency virus (HIV), among others [35,36], and has been associated with impaired immune responses [27,37,38,39]. An association between ROS, such as NO, the superoxide-radical (O_2_^−^), and peroxynitrites, and endothelial damage and inflammation has been reported [40]. Both endothelial damage and inflammation appear to play a vital role in COVID-19 [41].

It should be noted that a close relationship between inflammation and oxidative stress exists. High production of free radicals at the site of infection by immune cells, especially macrophages, triggers oxidative stress. Excessive extracellular ROS/RNS, characterized by malondialdehyde (MDA), 8-hydroxy-2′-deoxyguanosine (8-OHdG), and isoprostane accumulation [31,42], can either oxidize biomolecules including RNA/DNA, lipids, or proteins, or can structurally modify proteins and genes to trigger signaling cascades that can lead to the onset of the inflammatory response. The recognition of detrimental stimuli is initiated by pathogen-associated molecular patterns (PAMPs) [43], in immune cells and non-immune cells, via triggering of germline-encoded pattern-recognition receptors (PRRs), [44,45]. Inflammatory stimuli can trigger the activation of intracellular signaling pathways involved in the expression of inflammatory mediators. Primary inflammatory stimuli cause the release of microbial products and cytokines, including interleukin-1β (IL-1β), interleukin-6 (IL-6), and tumor necrosis factor alpha (TNF-α). These mediate inflammation through the activation of receptors, for instance toll-like receptors (TLRs), IL-1, IL-6 receptors, and the TNF-receptor [46]. As a result, intracellular signaling pathways are stimulated, including the mitogen-activated protein kinase (MAPK), nuclear factor kappa-B (NF-κB), Janus kinase (JAK)-signal transducer, and activator of transcription (STAT) pathways [47,48,49].

Overall, at first exposure to pathogens, a strong response of the innate defense system occurs at the beginning of the infection phase [50]. As such, a certain level of inflammation is physiological and required for optimal triggering of the immune response. However, low-grade systemic inflammation is common in several conditions, including cardiovascular disease (CVD), inflammatory bowel diseases (IBD), diabetes type 2 (T2D), arthritis, cancer, and obesity [51].

Individuals with low-grade chronic inflammation present a dysregulated innate immune system, resulting in an increased risk of infection [52]. Several issues may arise during extreme conditions as those triggered by COVID-19 [53]. In fact, complications related to the severe acute respiratory syndrome caused by SARS-CoV or SARS-CoV-2 are mainly due to pronounced inflammation caused by viral replication [54]. The infection causes increased secretion of IL-1β, IL-4, IL-10, interferon gamma (IFN-γ), IP-10, and monocyte chemoattractant protein 1 (MCP-1) [55]. In addition, it has been noted that human bronchial epithelial cells respond to SARS-CoV infection by producing several NF-κB-mediated cytokines, including IL-6 and IL-8 [56]. Also, patients with severe COVID-19 in intensive care units have shown elevated plasma levels of several cytokines (IL-2, IL-7, IL-10, TNF-α), MCP1, granulocyte-colony stimulating factor (GCSF), IFN-γ-induced protein 10 (IP-10), and macrophage inflammatory proteins (MIP-1A), suggesting a cytokine storm, resulting in hyper-inflammation and excessive reactions, such as hyperpyrexia and organ failure associated with disease severity, which can be life-threatening [53]. In fact, the severe COVID-19 associated cytokine profile (described also as macrophage activation syndrome (MAS) or secondary hemophagocytic lymphohistiocytosis (sHLH)), is characterized by hyper-inflammation together with multiorgan failure [57,58]. Furthermore, the cytokine storm response has been associated with an increased mortality risk, mainly due to respiratory failure from acute respiratory distress syndrome [59].

In addition to the activation of the inflammatory response and promotion of oxidative stress that are intimately related with the immune system [60,61], the immune system is interrelated with multiple aspects of physiological regulation such as hormonal regulation [62,63], metabolic regulation [64,65,66], circadian rhythms [67,68], as well as nutrient utilization [69]. Overall, malnutrition can compromise the immune response [70], altering cell regeneration and function and making individuals more prone to infection [71]. For example, it is well known that in economically poor areas with protein-malnutrition [72], the risk of infection is elevated, though additional factors such as tropical diseases may also be a factor [73,74]. In addition, inflammation related to poor dietary habits has reached alarming proportions [75], especially with regard to numerous cardio-metabolic diseases related to low-grade chronic inflammation such as overweight/obesity, T2D, metabolic syndrome, and auto-immune and cardiovascular diseases. An increasing body of evidence highlights that dietary modifications and nutritional factors can strongly impinge on chronic low-grade inflammation [76,77] and viral infection risk and symptoms (Table 1). These can serve best via the prevention of inflammation and oxidative stress and strengthening the immune system, but may also aid in the management of disease and infections as an adjuvant strategy. Consideration of dietary constituents and nutritional factors during the COVID-19 crisis can serve to strengthen the immune system for infection prevention, as well as to promote overall health in times of ongoing national restrictions. The following sections review the roles of various dietary constituents in infection, inflammation, and oxidative stress.

## 3. Dietary Constituents as Key Factors of a Strong Immune System and Low Infection Risk

### 3.1. Macronutrients

#### 3.1.1. Proteins

Low protein status due to low protein intake, i.e., below the recommended 0.8 g/kg body weight as proposed by the recommended dietary allowance (RDA), [101], such as in economically challenged countries with low protein availability, has been well recognized to increase the risk of infection [15]. It is believed that the low pool of available proteins also results in a decreased amount of functional active immunoglobulins and gut-associated lymphoid tissue (GALT), which play a role in gut-mucosal defense against infection [102], among others. Very low protein intake (2% of energy) increased the severity of influenza infection in mice, for example, through a low antibody response, and enhanced virus persistence in the lungs, related to hyper-inflammation and associated mortality [79]. Even if the prevalence of protein energy malnutrition (PEM) is low in Westernized countries, several protein sources, especially from food items such as processed meats, and cheese, are high in calories and saturated fats and can aggravate post-prandial effects, favoring lipogenesis and elevated inflammation [103,104]. In this respect, the rather pro-inflammatory aspects of proteins from animal sources and anti-inflammatory properties of plant derived proteins have been acknowledged [105]. For example, meat-protein-rich diets increase colonic monocytes [106], though it can be assumed that other matrix components such as saturated fats do play a role as might the absence of fiber and further phytochemicals.

Thus, protein intake of high biological value and from healthy dietary choices, such as from eggs, fish, lean meat (e.g., poultry), and whey protein (or other non-fat dairy protein), when eaten together with meals, may lower post-prandial lipogenesis and inflammation [107]. It is also known that proteins of high biological value, i.e., containing the essential amino acids in required amounts, can reduce post-meal glycemic response and improve satiety due to their effect on prolonged gastric-retention and gastro-intestinal transit time [108,109]. Therefore, high quality proteins are an essential component of an anti-inflammatory diet [110]. The consumption of a certain amount of proteins of high biological value is known to be crucial for optimal production of antibodies [111]. Branched-chain amino acids may maintain villous morphology and increase intestinal immunoglobulin levels, thereby enhancing the gut barrier and response [112]. Some amino acids modulate the metabolism and immune functions [113,114]. For instance, arginine supplementation increased the response of T-lymphocytes and T-helper cell numbers, and rapidly returned to normal T-cell function after operations, compared to control subjects [114], suggesting a role in prolonged or repeated infection.

Glutamine is required for the expression of a variety of genes of the immune system [113,115,116,117]. Glutamine is an energy substrate for macrophages, neutrophils, and lymphocytes, needed for pathogen-identification through the proliferation of immune cells and the repair of tissues [118]. For instance, in the immune system, glutamine plays a key role in controlling the proliferation of cells such as lymphocytes, neutrophils, and macrophages [113,116], activating proteins partaking in signal transduction, e.g., ERK and JNK kinases. Both can activate a number of transcription factors, including JNK and AP-1, eventually promoting the transcription of genes participating in cell proliferation [113,119,120]. In addition, a sufficient glutamine level is important for expressing key lymphocyte cell surface markers and also various cytokines, for instance IL-6, IFN-γ and TNF-α [116,117,121,122].

From observations in humans and from experiments in animals, it is known that a diet with a very low content of protein can be detrimental to fighting off infection. For example, mice fed a 2% versus a 20% protein diet, rapidly died following exposure to *M. tuberculosis* [80]. This was related to a reduced expression of IF-γ, TNF-α, and iNOS [80]. Interestingly, these effects were reversed rapidly within 2 weeks of changing the diet [123]. In humans, protein malnutrition and increased susceptibility to Zika and influenza viruses is related to cell mediated immunity and decreased bactericidal function of neutrophils, the complement system, and IgA as well as antibody response [124]. Low protein status, characterized by low albumin or pre-albumin levels, but also low iron and vitamin E correlated with lower responses to influenza vaccination in the elderly, thereby highlighting the interrelation between various nutrients and the immune response [78].

#### 3.1.2. Lipids

Fatty acids (FAs) may significantly alter immune responses, including changes in the organization of cellular lipids and interactions with nuclear receptors [125]. FAs in particular have been shown to affect the homeostasis and immune cell functioning in mice, e.g., epithelial cells, macrophages, dendritic cells, innate lymphoid cells, neutrophils, and T- and B cells [126]. In general, increased fibrinogen and high-sensitivity C-reactive protein (hs-CRP), an acute phase protein of hepatic origin, have been linked to saturated FAs consumption, while lower hs-CRP levels have been linked with polyunsaturated FA [127]. In particular, omega-3 FAs appear to have the most potent anti-inflammatory capability, though not all omega-3 FAs are anti-inflammatory [18]. Trans-fatty acid intake, especially from processed foods such as fries and chips, has also been described as pro-inflammatory, being associated with increased TNF-α, IL-6, and hs-CRP levels [12,128].

The two essential FA classes, omega-6 and omega-3, need to be consumed within the diet as the human body is unable to produce them. The intake of omega-3 FAs from fish and seafood has been shown to trigger anti-inflammatory reactions via oxygenated metabolites (oxylipins), including resolvins and protectins [129,130]. Omega-3 FAs include α-linolenic acid (ALA) consumed from various plant sources (Table 2) and eicosapentaenoic acid (EPA) and docosahexaenoic acid (DHA) consumed especially from fish and seafood sources, such as salmon, mackerel, and tuna. Omega-6 FAs such as arachidonic acid are primarily pro-inflammatory [75,131,132], constituting precursors of several pro-inflammatory mediators including eicosanoids (a subset of oxylipins) such as prostaglandins and leukotrienes derived from cyclooxygenase (COX) and lipoxygenase (LOX) enzymes, respectively [129]. Eicosanoid signaling resembles that of cytokine signaling and is part of the innate immune response [133]. Their signaling is influenced by phospholipase A2 activity, which regulates the different phases of the inflammatory response by releasing eicosanoids. Furthermore, their regulation includes class switching in which pro-inflammatory eicosanoid synthesis (and accompanying TLR4 activation of NF-κB induces pro-interleukin-1B) switches to anti-inflammatory (eicosanoid and docosanoid) synthesis. Many omega-3-derived oxylipins are specialized pro-resolving mediators, capable of enhancing bacterial clearance together with a downregulation of pro-inflammatory cytokines and enhanced removal of apoptotic neutrophils [132]. This is believed to contribute to the positive health effects of omega-3 consumption.

An imbalance of FA, such as saturated/unsaturated FAs, and omega-6/omega-3 FAs has important implications for immune system homeostasis, which can foster the onset of allergic, autoimmune, and metabolic conditions [126,134,135,136,137]. As omega-3 and omega-6 FAs may compete for the same enzymes, elevated omega-6 FA concentrations can hamper the omega-3 FA metabolism [138]. Thus, it is recommended to keep a healthy balance between omega-6 and omega-3, with a ratio of 1:1–4:1 [139,140]. Unfortunately, the intake-ratio of omega-6 to omega-3 FAs has been reported to be in the range of 10:1 in individuals consuming Westernized diets, thereby possibly fostering pro-inflammatory responses [141]. For example, in patients with rheumatic diseases, characterized by strong chronic inflammation, omega-3 fatty acid administration resulted in improvements of eicosanoid synthesis toward a less inflammatory profile and reduced pro-inflammatory cytokine production [142]. Responses following 4-month supplementation of omega-3 fatty acids of 2.5 g/d and 1.25 g/d vs. placebo were compared in a randomized controlled trial (RCT) study in healthy adults. A significant decrease in serum IL-6 by 10% and 12% in the low and high dose omega-3 FAs groups, respectively, was seen, compared to a 36% increase in the placebo group [143]. Likewise, another RCT reported that omega-3 FA supplementation resulted in reduced IL-6 production, while decreasing omega-6/omega-3 ratios reduced the production of stimulated IL-6 and TNF-α [144].

These anti-inflammatory effects are expected to contribute to a better immune-system. Indeed, in a study with mice, the omega-3 FA-derived mediator protectin D1 reduced viral replication, and improved survival and symptoms following influenza infection [81]. On the other hand, care has to be taken; in an earlier study, mice fed fish oil for 2 weeks had a reduced inflammatory state, even in the lungs, that resulted in increased morbidity and mortality, the latter by 40% [145]. This was accompanied with reduced CD8^+^ T cell populations and decreased mRNA expression of protein1-α, TNF-α, and IL-6. Thus, while reducing inflammation during hyper-inflammation and cytokine storm conditions is likely beneficial, a general reduction of inflammatory state during infection may be a double-edged sword. In another study with mice, consumption of 6 weeks of omega-3 FA-rich diets had no effect on vaccinia virus infection of the respiratory tract [146]. It can be hypothesized that prior FA status, amount, time of omega-3 FA intake, and state of infection need to be considered. Unfortunately, well-designed human studies on this are missing.

In addition to omega-3 FA, the amount of lipid-intake has been discussed as playing a role in viral infections. In mice, diets rich in lipids/fat seem to play a crucial role in both respiratory and extra-respiratory complications of influenza A virus infection, related to an increase in viral load in the lungs and heart. This deficient antiviral response has been associated with signaling defects in the inflammatory response in mice, leading to high lung inflammation and damage, as well as increased heart inflammation and damage, i.e., increased left ventricular thickness and mass [82]. A high-fat diet administration in mice has also been associated with an efficacy reduction in influenza vaccine through a reduced antibody response, due to macrophage dysfunction in fatty environments [83,147].

#### 3.1.3. Carbohydrates and Dietary Fiber

High glycemic index-induced acute hyperglycemia and acute insulin response, due to high consumption of processed carbohydrates (white flour, refined sugar), lead to an overload of the mitochondrial capacity and an increase of the production of free radicals [110]. Even a single high glycemic index meal has been associated with an immediate increase of inflammatory cytokines and C-reactive protein [148,149]. Increased levels of TNF-α and IL-6 have also been correlated with a higher glycemic index/glycemic load GI/GL [150]. Even if isocaloric, choosing higher-quality carbohydrates can improve postprandial glycemia and lower inflammatory responses [151]. In contrast, less processed, low-GL foods, such as vegetables, fruit, nuts, seeds, and whole grains, do not trigger such adverse post-prandial inflammatory effects [152]; this is attributed to more complex food matrices slowing down the digestion and absorption of carbohydrates. Interestingly, the administration of a ketogenic diet, i.e., a high fat but low-carbohydrate diet (<10% energy), seemed to protect mice from the severity of influenza A virus infection in terms of morbidity and mortality through the expansion of gamma delta T cells in the lungs. These cells play an essential role of host defense against the influenza A virus infection [84].

Dietary fibers are mostly complex carbohydrates and are an important factor regarding the influence of carbohydrates on inflammation [153,154]. A significant reduction in hs-CRP concentrations has been observed with increased fiber (i.e., 3.3 g/MJ or approximately 30 g/d) consumption [13]. In general, while the intake of 25 g and 38 g of fiber for women and men, respectively, is recommended, true intake is generally lower (around 15–20 g/d), at least in Westernized countries [155]. Another advantage of whole-grain intake is also a more favorable gut microbiome composition, which lowers both gut and systemic inflammation, and even small increases of only 5 g additional fiber per day can be beneficial [156,157]. Increased whole-grain intake (again with amounts of fiber even below 5 g/d) has been associated with decreased hs-CRP, IL-6, and TNF-α and increased SCFA [11,158,159,160], markedly decreasing inflammation-mediated disease risk [161,162], such as CVD, T2D, cancer, and obesity.

Different types of dietary fiber are not the sole influencers of gut health, but they do have a major effect. SCFA production, influencing gut microbiota species toward a composition perceived as healthier and strengthening gut mucosal integrity may be the most important aspects in this regard. SCFA levels are regulated by the diet, and given their importance in their immunomodulatory functions, their intake has been very much discussed in relation to inflammatory diseases in Westernized countries [163,164]. SCFAs such as acetate, propionate, and butyrate constitute important fatty acids that are produced by the gut microbiota during dietary fiber fermentation [165]. In cancer, intestinal homeostasis has in particular been attributed to the SCFA-related histone deacetylase inhibition, involved in enhancing the inflammatory response through gene regulation of cell proliferation and differentiation [166,167,168,169,170]. Anti-inflammatory signaling cascades are also activated through the SCFA-related G-protein-coupled receptor (GPRs) activation, e.g., GPR109A, GPR41, and GPR43 [171,172,173,174,175,176]. Indeed, SCFAs exert anti-inflammatory effects via IL-12 inhibition and upregulated IL-10 production in monocytes [177], repressing pro-inflammatory molecules release such as TNFα, IL-1, and NO [178], and reducing NF-ĸB expression [178,179]. SCFAs might be key regulators of inflammatory diseases by tightly controlling the migration of immune cells toward inflammatory sites as well as by modulating their activation state, enabling accelerated pathogen clearance through ROS activation [163,164].

In addition to SCFA, dietary fiber has been reported to increase the diversity of gut microbiota and promote health-associated bacteria such as *Bifidobacterium* spp. and *Lactobaccillus* spp., which have been related to, among others, mucosal inflammation [180]. Such species could contribute to reducing the growth of health-detrimental pathogens, including *Clostridium* spp. [181]. A healthy gut microbiota rich in *Bifidobacterium* spp., *Faecalibacterium* spp., *Ruminococcus* spp., and *Prevotella* spp. has been associated in a systematic review with lower systemic inflammation, characterized by reduced hs-CRP and IL-6 [182], even though probiotic interventions with bacteria have not always shown systematic benefits [183]. Furthermore, dietary fiber may, as has been shown in a mouse model, also enhance mucosa thickness. This may prevent bacteria from degrading this important barrier, through which allergens and other microbes could otherwise infiltrate the human host [87]. For further interactions between the gut microbiota, diet, and health-related aspects, the reader is referred to more comprehensive reviews [184,185].

It is important to highlight the emerging role of the microbiota, its modulation by nutrition, and its influence on responses to viral infection, though human studies linking diet, gut microbiota, and infection are scarce. While mainly the gut-microbiota has been studied in relation to the immune system, the nasopharyngeal microbiota may be involved in the etiology of respiratory infections [186]. An interplay between respiratory tract infections and the gut microbiota has been emphasized. Although viral infections can change the microbiome, the latter also is involved in adaptive immune responses against respiratory pathogens [187,188], triggering immune reactions of the innate system [189]. For instance, responses of macrophages to respiratory viruses are linked to the presence of distinct gut microbes. The importance of the gut microbiota for functioning innate immune responses has been demonstrated in animal models. Antibiotic treatment of animals resulted in defective macrophage responses to IFNs, resulting in hampered effects to control viral replication [190].

Health promoting effects of intestinal microbiota against viral infections, including influenza, have been known for more than a decade [191,192]. It has become clear that these effects depend on immune-regulatory cells. In particular, elevated mRNA levels, i.e., IL-1β, IL-2, IL-21, IL-18, IL-12, and IL-15, circulating through the lymphatic and circulatory systems (gut-lung axis) have been observed in mice pre-treated with probiotics. Also, studies employing mouse models indicated the microbiota contributed to establish a defense-system against pathogens, such as blocking cell internalization [193], binding to and destabilizing virion morphology [194], inhibiting further influenza virus infections [195], but also suppressing other viral replication, i.e., of herpes simplex virus (HSV)-2 [196]. It is obvious that the gut microbiota forms a dynamic environment that can be disturbed by virus infection, but can be positively modulated by dietary constituents. COVID-19 has been associated with both respiratory and gastroenteritis symptoms [197]; the latter can affect the diversity of the gut microbiota and increase the risk of contracting secondary bacterial infections.

Interestingly, dietary fiber consumption in adult Americans has been inversely linked to the risk of death from respiratory and infectious diseases [85]. In this study, for each 10 g increase in dietary fiber per day, the mortality-relative risk from infectious and respiratory diseases decreased by 34% and 18% in men and 39% and 34% in women, respectively. In another observational study with 11,897 US men [198], dietary fiber intake was associated with reduced risk of chronic obstructive pulmonary disease (COPD, not necessarily related to infection). Highest versus lowest quintiles showed an odds ratio (OR) of 0.85 for total fiber, 0.83 for cereal fiber, and 0.72 for fruit fiber, all of which were significant. Of note, T-cells from the lungs may migrate to the gut to produce type I IFNs. This may then result in Th17 cell activation, increasing IL-15 and IL-17 secretion, damaging epithelial cells in the gut [199].

The importance of both prebiotics and probiotics for infection risk prevention have just recently been highlighted [200]. In a recent systematic review including 11 RCTs with almost 2500 children, it was highlighted that probiotic treatment reduced respiratory infections [201]. As dietary fiber can influence gut microbiota and have prebiotic functions, it may be hypothesized that a similar effect can be found also with dietary fiber, though direct evidence for humans is, to our knowledge, lacking. Additional evidence has emerged from animal models. Recently, it was shown that mice fed dietary fiber exhibited increased survival rates to influenza virus infection [86] by increasing macrophages with a reduced production of the chemokine CXCL1 (causing neutrophil recruitment to the lung), as well as improving CD8^+^ T cell function. In another study, mice on a fiber-free diet were more susceptible to *Citrobacter rodentium*, a mucosal pathogen, likely due to the mucosa erosion also observed [87].

How to modulate the gut microbiome in the context of the COVID-19 crisis through dietary fiber and other dietary constituents corresponds to an important research aspect to be considered in future studies.

### 3.2. Micronutrients

Micronutrients encompass 12 vitamins, as well as several macro-minerals and trace elements, all of which are defined as essential. By contrast, the roles of ultra-trace elements, such as nickel or boron, are unclear. A decreased status of micronutrients often causes more or less specific deficiency symptoms. However, the cut-off points for deficiency may be unclear, and also depend on individual variability. Furthermore, it is often debatable how best to define micronutrient status, though concentrations in blood generally serve as the best available proxy marker [202], with a few exceptions such as for iron (measured by hemoglobin). However, plasma and serum are often a fast-exchanging pool, and levels in tissues would be a superior marker for status determination.

#### 3.2.1. Vitamins

##### Vitamin A

Vitamin A deficiency has traditionally been associated with increased risk of infection [203,204]. In fact, it is among the most abundant micronutrient deficiencies worldwide, especially in countries with low protein and meat intake [205,206]. However, vitamin A can also be formed out of pro-vitamin A carotenoids such as α- or β-carotene, being the major vitamin A source in individuals with low meat intake [207,208]. Vitamin A is important for the morphology of the epithelium, playing a role in its keratinization, stratification, differentiation, and functional maturation [209], constituting a front line of defense against pathogens. Vitamin A is involved in the formation of healthy mucus layers, such as those of the respiratory tract and the intestine, required for mucin secretion and enhancing antigen non-specific immunity functions [209,210].

Retinal, retinol, and retinoic acid are active forms of vitamin A, and the latter acts as a ligand, activating the nuclear retinoic acid receptor (RAR) and unknown metabolites may activate the retinoid X receptor (RXR) [211]. Therefore, (*all-trans and 9-cis*) retinoic acids play crucial roles in the regulation of the differentiation, maturation, and function of the innate immune system and cells, e.g., macrophages [212] and neutrophils [213]. Retinoic acid promotes an immediate response to pathogen invasion by phagocytosis and activation of natural killer (NK) T-cells, which relate immune-regulatory functions through cytotoxic activity [214,215]. Retinoic acid can also alter the differentiation of dendritic cell precursors [216,217,218]. These are specialized sentinels of the immune system, orchestrating innate and adaptive immune responses [219]. It is thus not surprising that low vitamin A status (measured typically as serum retinol) has been shown to correlate with hampered function of neutrophils, macrophages, as well as T-and B-cells [220]. A critical role in the etiology of influenza, together with vitamin D, has been also proposed [219]. Furthermore, individuals with low vitamin A status exhibit histopathological alterations to the pulmonary epithelial lignin and lung parenchyma, resulting in increased risk of lung dysfunction and respiratory disease [221]. This is particularly relevant considering the effects that COVID-19 has on lung function [222].

It appears that direct clinical evidence linking vitamin A administration to enhanced resistance to infection or proxy markers such as cytokine production or lymphocyte activation is lacking [223]. Furthermore, a lack of association between plasma β-carotene, retinal (as well as α-tocopherol and zinc), and response to the influence of a vaccine was shown in the elderly, suggesting that the immune response was not significantly influenced by the differences of these micronutrients in this population [224]. This may be because a low status of vitamin A in Westernized countries is rather uncommon. Of note, vitamin A supplementation did not affect the risk of lower respiratory diseases (LRD) and symptoms in a systematic review in children [88], suggesting that vitamin A supplementation should not generally be recommended for LRD prevention. Similar results were found in an earlier meta-analysis, where risk of infection of acute respiratory diseases in developing countries was non-significantly higher in the supplemented group [89]. In a recent review, it has been highlighted that vitamin A losses occur in infection via impaired vitamin A absorption and urinary losses [19]. In general, studies investigating the efficacy of vitamin A supplementation on improvement of immune responses to vaccines have produced conflicting results [225]. However, it is suspected that the influence of pre-existing vitamin A stores and extra-nuclear activities of their hormone receptors play crucial roles in resulting responses. More well-controlled studies are warranted in order to sufficiently investigate this topic.

##### Vitamin D

Vitamin D can be taken up from the diet via fish, eggs, fortified milk, and mushrooms, but it can also be synthesized under the skin in the presence of UV-light from cholesterol. The active form of vitamin D, calcitriol (1,25 dihydroxyvitamin D), formed following kidney and liver hydroxylations, is most renowned for its regulating role in calcium homeostasis and thus bone health, but it has also been shown to regulate the immune system [226]. In fact, T-cell functioning is closely related to vitamin D [227]. T-cells express the CYP27B1 gene, responsible for processing the pre-hormone vitamer of vitamin D (25-hydroxyvitamin D), calcidiol into the active hormone (calcitriol). Only after binding to calcitriol can T-cells can carry out their physiological functions [228]. Other immune cells are involved in CYP27B1 expression, e.g., macrophages and dendritic cells, enabling vitamin D activation [227,228]. As for vitamin A with RAR/retinoid X receptor (RXR), the activated vitamin D form of calcitriol can bind to a specific nuclear receptor (vitamin D receptor, VDR). While this receptor is especially known for its role in regulating phosphorus and calcium levels and thus bone metabolism, its role for both the innate and adaptive immune systems has been highlighted [229]. Vitamin D has been controversially discussed for its role in influenza prevention and therapy [230], and the WHO has discussed its role in the prevention of respiratory diseases in children [91].

Regarding human trials, mixed effects have been reported. In a study in China, protective effects of vitamin D administration were reported regarding incidence and severity of influenza [231] between low- and high-dosed vitamin D children, though low vitamin D status at study onset could have played a role. In a recent review discussing intervention trials, mixed results were observed [230]. Vitamin D therapy, in a meta-analysis, improved conditions in individuals with chronic obstructive pulmonary disease (COPD), though this was not caused only by infection [232], similar to another meta-analysis [232]. Another review reported on a reduced risk of influenza and COVID-19 infections and mortality [233], mostly due to related inflammatory status and anti-microbial peptides such as cathelicidin and defensins and by modulating adaptive immunity, such as reducing Th1 helper cell responses. This is corroborated by a meta-analysis of RCTs, finding protective effects of vitamin D against respiratory tract infection, with daily dosing appearing to be the most effective strategy [92]. Other, more direct effects on virus-receptor binding could also play a role. Interestingly, vitamin D supplementation promoted binding of the SARS-CoV-2 cell entry receptor ACE2 (angiotensin converting enzyme 2) to AGTR1 (angiotensin II receptor type 1), reducing the number of virus particles that could attach to ACE2 and enter the cell [26].

A recent retrospective study including 780 confirmed cases of SARS-CoV-2 infection determined mortality and associated factors, with a special focus on vitamin D status. Older and male cases with pre-existing conditions and below normal vitamin D levels were strongly associated with increasing odds of death, those with insufficient vitamin D status were almost 13 times as likely to succumb [90].

Moreover, recent observational studies investigating COVID-19 infection and mortality globally have observed a latitude effect with most Southern Hemisphere countries (except for Brazil), reporting less cases/mortality [234]. Interestingly, at the same time, the Southern Hemisphere is entering autumn and will have the highest vitamin D blood levels at this season. Conversely, Northern Hemisphere countries are entering Spring and have their lowest blood vitamin D levels after winter. In Europe, COVID-related mortality appears blunted with increased latitude. For instance, Nordic countries such as Finland and Norway that have either mandatory vitamin D fortification or higher vitamin D intakes have among the highest vitamin D levels in Europe but also lower mortality. Conversely, despite the high sun exposure, older populations in Italy and Spain show a much lower vitamin D status and higher COVID-mortality rates [235]. Surely, correlations are not causation (as infection protocols, testing, population isolation, healthcare, etc. all modify infection rates), but it provides an interesting research hypothesis to confirm with RCTs.

##### Vitamin E

Vitamin E exists in the major forms of tocopherols and tocotrienols, with most research focused on the effects of the former. Tocopherols are present in high amounts in nuts and vegetable oils while tocotrienols are found predominantly in some seeds and grains. Although deficiencies in vitamin E are uncommon in humans, secondary deficiencies might occur, for example, following an intestinal malabsorptive disorder. Of note, in order to replicate its antioxidant effects, vitamin E works synergistically together with vitamin C, by which its tocopheroxyl radical is reduced by vitamin C [236].

Vitamin E has also been shown to regulate the maturation and functions of dendritic cells [237], which are important for intertwining innate and adaptive immune systems to orchestrate immune response [238,239,240]. In addition to increasing NK cells activity, by modulating NO levels [241], vitamin E administration enforces humoral (B cells) as well as antibody responses, both in animals and humans [242,243]. Vitamin E has been shown to improve naïve T-cells immune synapse formation and initiate T-cell activation signals [243,244,245,246].

The role of vitamin E in the prevention of infections such as influenza has been discussed [247,248,249], but well controlled studies in humans are lacking. In a study with mice, vitamin E administration (60 mg/kg per day for up to 7 d) was superior in reducing the elevated oxidative stress resulting from influenza infection [248] compared to vitamin C (80 mg/kg), but the combination of both was most successful in reducing lipid peroxidation. Following influenza infection in mice, vitamin E supplementation decreased lung related-pathology and mortality through the improvement of the T-helper 1-type cytokines response, which produces pro-inflammatory responses against intracellular parasites [250].

Supplementation in humans with vitamin E seems to restore IL-2 production, improving T-cell proliferation and immune system functioning [251,252]. A recent study conducted in Malaysian adults in which volunteers received either tocopherol or tocotrienol supplementation showed increased expression of a variety of genes associated with the immune response [253]. Interestingly, the specific genes altered were different between the two groups. Furthermore, vitamin E supplementation, after a first hospitalization in elderly with pneumonia, was associated with 63% reduced re-hospitalization within 90 days [254]. In an RCT, elderly, healthy participants receiving 200 mg/d capsules of vitamin E for 4 months had a 65% increase delayed-type hypersensitivity skin response (*p* = 0.04) compared to placebo, and also improved antibody titers of hepatitis B and tetanus vaccines, highlighting implications in T-cell mediated functions [243]. In another study conducted among 2216 smokers receiving 50 mg/d of vitamin E for 5–8 years, it was shown that vitamin E supplementation reduced pneumonia incidence by 69% in elderly men [93].

##### Vitamin C

Vitamin C is often perceived as a classical antioxidant, directly quenching free radicals in the aqueous layer while being oxidized itself to dehydro-ascorbic acid. In addition, an increased dietary ascorbic acid intake has been related to lower concentrations of C-reactive protein and tissue plasminogen activator [20]. However, vitamin C also acts as a cofactor for a number of biosynthetic and gene regulatory monooxygenase and dioxygenase enzymes, suggesting immune-modulating effects [255,256,257,258,259]. Several in vitro (cell-culture) and pre-clinical studies have highlighted the barrier-enhancing effects of vitamin C, especially regarding lipid synthesis, via influencing signaling and biosynthetic pathways [260,261,262,263,264]. In addition, ascorbic acid can alter gene expression in dermal fibroblasts, enhancing their proliferation and migration that play paramount roles for tissue remodeling, important in wound healing, for example [265,266]. Vitamin C has been shown to stimulate the migration of neutrophils to the infection site, stimulating phagocytosis and ROS generation [267,268,269].

In parallel, vitamin C can also stimulate neutrophil apoptosis, protecting host tissue from strong damage [270], and it further aids in macrophage removal [268]. Finally, ascorbic acid plays a role in the differentiation and maturation of T-cells [271,272]. Similar maturation has been observed with immature NK cells, as well as proliferative and differentiation effects with mature NK cells [273].

Low vitamin C status has been discussed as an adjuvant measure to aid in individuals with the common cold and also pneumonia [274], and positive effects were found in some intervention trials such as shortening the duration of colds [274]. In a study with mice exposed to restraint stress and virus-induced pneumonia (H1N1) [275], vitamin C administration (125 and 250 mg/kg) was shown to reduce mitochondrial antiviral signaling (MAVS) and interferon regulatory factor 3 (IRF3) and to increase NF-κB expression, while reducing steroid hydroxylating enzymes. As reviewed earlier [276], a few controlled studies found significant benefits for supplementing vitamin C in subjects with pneumonia. For example, in a double-blinded controlled trial with elderly participants, 200 mg/d of ascorbic acid for 4 weeks improved respiratory condition [277]. In a recent meta-analysis of nine RCTs, extra doses (0.7 to 8 g/d) of vitamin C against common cold virus infections reduced duration of infection, shortened time of indoor confinement, and relieved symptoms [94]. In another meta-analysis of eight RCTs in 3135 children, supplementation of vitamin C with 0.5–2 g/day did not prevent infection of upper respiratory tract diseases, but reduced the duration of infection by 1.6 days [278].

##### B Vitamins

B vitamins are involved in many energy-related enzymatic processes. Riboflavin (vitamin B_2_), as it is a photosensitizer, was employed together with ultra-violet radiation to reduce viral load of batches of blood for transfusion, and as such impinged effectively on the titer of the Middle East respiratory syndrome-related coronavirus (MERS-CoV) in human plasma to below the detection limit [279]. However, the effect of vitamin B_2_ alone is not known. Administration of vitamin B_3_ for lung injury treatment in mice, through nicotinamide treatment, significantly decreased inflammation and reduced the neutrophil infiltration, although hypoxemia was surprisingly increased [280]. NAD^+^-dependent enzymes and their role regarding inhibiting effects of nicotinamide deserve further investigation. Low plasma pyridoxal 5′phosphate (PLP), the active coenzyme form of vitamin B_6_, has been significantly associated with impaired humoral and cell-mediated immunity [281,282,283,284]. In critically ill patients, vitamin B_6_ supplementation increased plasma PLP concentrations associated with increased total lymphocyte cells, including T-helper and T-suppressor cells [285]. The inverse relation of vitamin B_6_ intake and inflammation status has been well reviewed previously [286].

In the case of folate (vitamin B_9_), there have been responses to high-dose folic acid supplementation on altered mRNA expression in cytokines, together with a lowered number of cytotoxicity of NK cells in healthy participants [287]. Individual studies have also shown that cobalamin (vitamin B_12_) may act as an immunomodulator. For example, vitamin B_12_-deficient patients have shown decreased levels of CD8^+^ cells, an abnormally high CD4/CD8 ratio, and subdued activity of NK cells [288]. Methylcobalamin administration to these patients improved the CD4/CD8 ratio and suppressed NK cell activity and CD3^-^/CD16^+^ cell increases [288]. Vitamin B forms have also shown to be efficient in lowering inflammation caused by virus infection. In particular, in patients with HIV, high vitamin B_3_, vitamin B_6,_ and vitamin B_12_ intake in the form of niacin, pyridoxine, and cobalamin, respectively, have been significantly associated with lower inflammation levels such as decreased CRP [95].

Prolonged periods (6 months) with high-doses (exceeding daily intake recommendations for adults) of B-group multivitamins was carried out within an RCT trial with 32 healthy adults and improved levels of total plasma homocysteine, as a marker of oxidative stress [289]. Homocysteine is associated with oxidative stress and is known to increase during B-vitamin deficiencies, especially folate and vitamin B_12_. In another long-term (7 years) RCT trial it was tested whether a daily administration of folic acid (2.5 mg), vitamin B_6_ (50 mg), and vitamin B_12_ (1 mg), versus placebo, could prevent CVD disease markers in women (n = 300). The combination treatment did not change major biomarkers of vascular inflammation [290]. Trials with a larger number of participants and stronger epidemiological designs would provide more definitive information about the role of B vitamins in the immune system and infection.

#### 3.2.2. Minerals

Reduced macro-minerals and trace elements have been associated with increased risk of infection. For instance, magnesium intake has been inversely associated with the concentrations of hs-CRP, IL-6, and TNF-α in a dose-dependent manner [9]. Magnesium is co-factor for many enzymes, playing a role in energy-metabolism, and a low status could interfere with numerous enzymatic reactions. Similarly, trace-elements such as zinc, copper, and selenium are required as co-factors for a variety of enzymes involved in antioxidant reactions, and have been implicated in boosting the immune system [291,292]. Several mineral deficiencies often occur together, such as low iron together with low zinc status [293].

##### Zinc

Zinc deficiency is a serious public health problem worldwide [294] and also appears to be prevalent in Westernized countries [295,296]. A low zinc status has been associated with increased risk of viral infections [297]. Zinc has been shown to be important for skin maintenance and mucosal membrane integrity [292], and the un-chelated free form of zinc has been shown to have direct antiviral effects, such as on rhinovirus replication in vitro [291]. Zinc is essential for cellular growth and differentiation of immune cells, having a fast differentiation and turnover rate, and helps to modulate cytokine release and trigger CD8^+^ T-cell proliferation [298]. Zinc has also been proposed to be crucial for intracellular binding of tyrosine kinase to T-cell receptors, which is needed for the development and activation of T-lymphocytes [299]. Zinc is an important cofactor for over 750 zinc-finger transcription factors, and thus implicated in DNA and RNA synthesis [300], also necessary for the production of immune-related proteins. By stabilizing the tertiary structure or as an essential component of the enzyme’s catalytic site [301], zinc serves as a cofactor for over 200 enzymes involved in antioxidant defense, notably of SOD and anti-inflammatory SMAD proteins [302].

In a recent review, the role of low zinc status in the elderly and its relation to pneumonia has been emphasized [303]. The mortality due to pneumonia has been reported to be twice as high in individuals with low zinc status versus individuals with normal zinc levels [303]. For some time, zinc has been suggested to improve symptoms of the common cold. In a randomized, double-blind, placebo-controlled study, patients (n = 100) with common cold symptoms took 13.3 mg of zinc as long as symptoms were present [96]. Compared to a placebo, zinc significantly reduced the duration of symptoms of the common cold, from 7.6 to 4.4 days.

##### Iron

Iron deficiency is highly prevalent worldwide [296], and its association with infectious diseases is well-recognized [304,305]. Often, low iron and low vitamin A status go together, as both are absorbed well from foods with a high content of protein such as meat and meat products [306,307]. Vitamin A appears to modulate hematopoiesis and iron metabolism, enhancing immunity to infectious diseases [306,307,308]. Iron appears as an essential component in cell differentiation, growth, and functioning (e.g., DNA synthesis through ribonucleotide reductase). Iron helps to fight off infections by enabling T-lymphocyte immune cell proliferation and maturation as well as regulating cytokines production and action against bacteria, for example, by neutrophil action [291,292].

The role of iron for bacterial infections [309] and viral infections [310], including respiratory infections [311], has been critically reviewed, highlighting that iron homeostasis and levels are tightly controlled. During inflammation, iron absorption is downregulated via hepcidin [309] in order to limit the available pool of iron for proliferating bacteria and virus particles and to limit excessive oxidative stress. However, during prolonged periods of iron deficiency, antibody production is typically reduced, as shown in experimental studies with mice exposed to influenza virus [97]. This has also been shown in elderly adults, relating iron deficiency to disturbed cell-mediated and innate immunity [312]. In a case-control study in 485 hospitalized 2–5 year old children receiving 3 months of iron supplementation, recurrences of acute respiratory tract infections, urinary tract infections, and gastroenteritis were significantly reduced [313].

##### Copper

Copper has been shown to have a role in the innate immune response to bacterial infections [292] and has been associated with IL-2 production and response. High concentrations of copper may be toxic for invading microbes and appear to be employed by macrophages as a defense strategy [314], which could play a role in secondary infections following viral infection. Copper is further involved in T cell proliferation [315], antibody production, and cellular immunity [298]. A healthy copper status has been related to aiding in the defense against several bacterial infections, including *E. coli, Salmonella*, and tuberculosis [316]. However, since copper needs are very low (it is often considered an ultra-trace element) and it is ubiquitously distributed, copper deficiency is rather rare.

##### Selenium

The role of selenium as an adjuvant therapy in viral and bacterial infections has been discussed [317], and its relationships with influenza virus, hepatitis C virus, coxsackie virus, among others, have been reported [317,318]. Low selenium status has been reported in multiple geographical regions including Western countries [319]. Requiring selenium for their synthesis, selenoproteins include several antioxidant enzymes such as GPx, selenoprotein P, and thioredoxin reductase [320,321]. Therefore, one of selenium’s primary roles is its ability as an antioxidant to quench ROS [322]. For example, these selenoproteins play vital roles in the hosts’ antioxidant defense system, influencing leukocyte and NK cell function [315]. It has been reported that selenium is protective against the heart-damaging effects of the cytomegalovirus [291] and is involved in T-lymphocyte proliferation and the humoral system, especially immunoglobulin production [291,315].

Selenium deficiencies have been associated with viral infections such as influenza, influencing adaptive and innate immunity responses and leading to a high level of virus-related pathogenicity. In this context, dietary selenium supplementations were suggested as adjuvant therapies of influenza infection, supporting the immune response [317]. Selenium supplementation seems to act on selenoprotein gene expression, improving the vaccine response [98]. However, supplementation with selenium may be a double-edged sword of low therapeutic width, as supplementation has been discussed in relation to elevated incidence of type 2 diabetes [323].

A prospective study was carried out in 83 patients with respiratory diseases requiring intensive care. Selenium levels in serum at admission were 28% lower in the intensive care unit (ICU) group vs. a general ward group. Poor serum selenium status was associated with decreased numbers of lymphocytes and albumin concentration, a marker of protein status, and correlated with increased CRP [324]. In a recent RCT, critically ill patients with acute respiratory distress syndrome received selenium in the form of sodium selenite (1 mg for 3 days and 1 mg/d for a further 6 days) [325]. Selenium concentrations were linearly associated with serum GPx levels and antioxidant activity, as determined by ferric-reducing antioxidant power. Both IL-1β and IL-6 serum concentrations were inversely associated with selenium serum levels. However, no effects on overall survival, the duration of mechanical ventilation, or ICU stay were apparent, which also raises caution in interpolating findings from sub-clinical markers to harder endpoints.

### 3.3. Phytochemicals

#### 3.3.1. Polyphenols

Individuals regularly consuming vegetables and fruits have lower rates of inflammatory markers such as CRP, IL-6, and adhesion factors [326,327,328]. This has been attributed to the high fiber content and higher concentrations of certain vitamins and minerals in these food items, together with a lower caloric density as described above, and other protective phytochemicals abundant in a plant-based diet. Adding fruits and vegetables to the diet, such as those rich in flavonoids, have significantly reduced serum inflammatory markers [21,329,330] by improving the microvascular reactivity, reducing CRP values [331,332], improving lipid profiles [333,334,335], and enhancing endothelial function [336,337,338]. A number of flavonoids including quercetin have been studied in vitro, i.e., in cell culture monolayers, to examine their potential antiviral properties, for example, infectivity and replication of herpes simplex virus type 1 (HSV-I), polio-virus type 1, parainfluenza virus type 3 (Pf-3), and respiratory syncytial virus (RSV). Quercetin decreased, dependent on its concentration, viral infectivity, and hampering of intracellular viral replication, when mono-layers were infected and subsequently cultured in quercetin-containing medium [339].

Among the most abundant phytochemicals or secondary plant compounds in the diet are polyphenols, with dietary intakes about 1g/d in Westernized countries [340]. The consumption of high-polyphenol diets, which likely exert numerous antioxidant [341,342] and anti-inflammatory effects [343,344] through inhibition of NF-κB and AP-1 and activation of Nrf2 [345], have improved lipid profiles and reduced dyslipidemia, via improving concentration and function of HDL-cholesterol and reduction in LDL-cholesterol [346,347,348]. In addition to their potential, direct antioxidant effects, several phytochemicals have been proposed to interact with transcription factors, especially NF-kB and Nrf-2 [22]. Polyphenols have also been advertised for their potential pre-biotic effects on the gut [349,350].

The role of polyphenols against influenza viruses, both regarding prevention and treatment, has been reviewed recently [351]. The major important mechanisms highlighted were suppression of neuramidase and hemagglutinin activity, influences on viral replication, viral hemagglutination, adhesion and penetration into the host cell, as well as modifying cellular signaling pathways and transcription factors. A strong anti-influenza virus activity in chicken embryo fibroblast cells and in mice was shown following the administration of an extract of *Geranium sanguineum* L., rich in polyphenols [352]. Likewise, coumarin derivatives (bis(triazolothiadiazinyl coumarin)) were studied for their anti-influenza activity. Within various cell models, the positive effect of coumarin against viral infections such as of HIV, influenza, enterovirus 71(EV71), and coxsackievirus A16 (CVA16) was shown. This was explained by different mechanisms, including changing protein conformation needed for viral entry, effects on replication and infectivity, and also cellular signaling regulation via AKT-mTOR (mammalian target of rapamycin), NF-κB, and Nrf-2, important for stimulating the body’s own antioxidative system [353]. In addition, theaflavin derivatives (polyphenols from black tea) had a strong inhibitory effect against influenza virus in vitro [354], perhaps due to downregulation of IL-6 expression. Similar anti-viral activities were reported for further polyphenol constituents due to the implications of MAPK kinases involved in the transport of viral ribonucleoprotein complexes in the cytosol and interactions in vital hemagglutinin protein via redox-sensitive pathways [355]. Though human RCTs, to our knowledge do not exist, an intervention trial in mice receiving a polyphenol extract from *Cistus Incanus* resulted in a lower influenza virus infection rate and reduced mortality of mice receiving the extract [99].

In a recent RCT, the effectiveness of eight weeks of curcuminoid supplementation, a rather apolar polyphenol, at a daily dose of 1 g, on markers of oxidative stress and inflammation in patients with metabolic syndrome was investigated. Supplementation with curcuminoids significantly improved serum SOD activity and reduced MDA and CRP concentrations compared with placebo [356]. The same study included a meta-analysis of data from all RCTs to study the impact of curcuminoid administration on CRP. Overall, curcuminoids vs. placebo reduced circulating CRP concentrations by about 2.2 mg/L.

#### 3.3.2. Carotenoids

Carotenoids are a group of mostly C-40 tetraterpenoid plant pigments and have been discussed for their antioxidant properties and quenching of ROS such as singlet oxygen and lipid peroxides within the lipid bilayer of the cell membrane [357]. Low levels of α- and β-carotene, lutein/zeaxanthin, as well as total carotenoids have been significantly associated with increased levels of oxidative stress, as recently reviewed [358], and also of inflammation [359]. For instance, low levels of α- and β-carotene, lutein/zeaxanthin, and total carotenoids were significantly more likely to be associated with increased IL-6 levels in women [359]. Another randomized, cross-over study determined the effect of orange and blackcurrant juice (high carotenoid content) supplementation in patients with peripheral arterial disease for four weeks; supplementation resulted in reduced inflammation markers, i.e., CRP and fibrinogen in plasma, while no significant differences regarding IL-6 and endothelial markers compared to placebo group were found [360]. Carotenoids such as lutein, zeaxanthin, and carotene plasma concentrations have also received attention for their potential antiviral roles [361]. A higher risk of mortality has been associated with low concentrations of plasma carotenoids in patients having contracted an HIV infection [100]. Carotenoids may also impact immune function by regulating membrane fluidity and gap-junction communication [362]. Finally, some carotenoids serve as precursors for vitamin A and may thereby exert immune-modulating functions attributed directly to vitamin A status.

A prospective serologic analysis of over 29,000 men revealed that higher serum β-carotene had significantly lower all-cause mortality, by about 19%. In addition, serum β-carotene was significantly associated with reduced risk of death from CVDs, cancer, diabetes, respiratory disease, and diabetes, providing evidence that higher β-carotene biochemical status is associated with a multitude of mortality causes [363]. However, cause and consequence could not be established in this study. In line with this finding are results from a meta-analysis in which one study highlighted that in elderly subjects, higher lutein/zeaxanthin levels were associated with 23% lower respiratory mortality [364]. Care should be taken as supplementation of higher doses of β-carotene (20–30 mg/d for several years) has caused increased cancer rates, though specifically in smokers and asbestos-exposed workers [365].

## 4. Summary and Future Directions

In summary, this review thoroughly covers nutrients, micronutrients, and phytonutrients known to affect immunity and infection risk, particularly relevant during the COVID-19 crisis. Figure 1 shows a schematic diagram illustrating interactions between selected dietary constituents, the immune system, and viral infection. Evidence indicates that a diet that positively impacts immune function contains adequate amounts of protein, particularly including glutamine, arginine and branched-chain amino acids (BCAAs); high omega-3 versus lower saturated, trans fat, and omega-6 fatty acids, low refined sugars, high fiber content such as whole grains, and micronutrients including vitamin A, vitamin D, vitamin C, vitamin E, B vitamins, zinc, selenium and iron, as well as phytochemicals. Table 2 shows examples of dietary sources. However, as shown well for dietary fiber, an excessive anti-inflammatory response may also reduce the immune response and increase susceptibility to infection. Controversial effects were also found for vitamin A, while carotenoid supplementation may only be detrimental for smokers and asbestos-exposed workers. Selenium supplementation should be considered with care due to the small therapeutic window and possible side-effects, especially in subjects with diabetes. While anti-inflammatory dietary constituents may be beneficial during hyper-inflammation, such as the COVID-19 cytokine storm, care should be taken against high doses of isolated anti-inflammatory compounds and/or antioxidants during healthier conditions in order to not suppress inflammation and the immune system too strongly.

The dietary components reviewed herein highlight the importance of an optimal nutrient status to reduce inflammation and oxidative stress, thereby strengthening the immune system during the COVID-19 crisis. The multiple associations highlight the complexity of the topic, and may also point out to not only individual marker determination, but rather toward the simultaneous determination of a nutrient signature, such as determined by nutrigenomics tools (i.e., metabolomics fingerprinting) [366]. Such an approach may also reveal yet unnoticed relations between nutrients, their metabolites, inflammation, oxidative stress, and the immune system. Furthermore, confounding factors such as medication and environmental pollutants could be also taken into account.

This work has been prepared with the purpose of contextualizing the available evidence to the current context of the COVID-19 pandemic crisis. An important additional aspect to keep in mind is the control of low-grade chronic inflammation related to chronic diseases such as obesity, diabetes, auto-immune, and CVDs. This might be possible through controlling nutritional deficiencies and promoting adequate nutritional status, which might improve immune response in infection phases. Our work was prepared with a focus on nutritional components available in the diet. However, some of these components are also available as supplements. Due to the potential limited therapeutic window of some of the dietary constituents discussed, we encourage obtainment of the beneficial nutritional and dietary constituents through the promotion of healthy dietary habits. 

## Figures and Tables

**Figure 1 nutrients-12-01562-f001:**
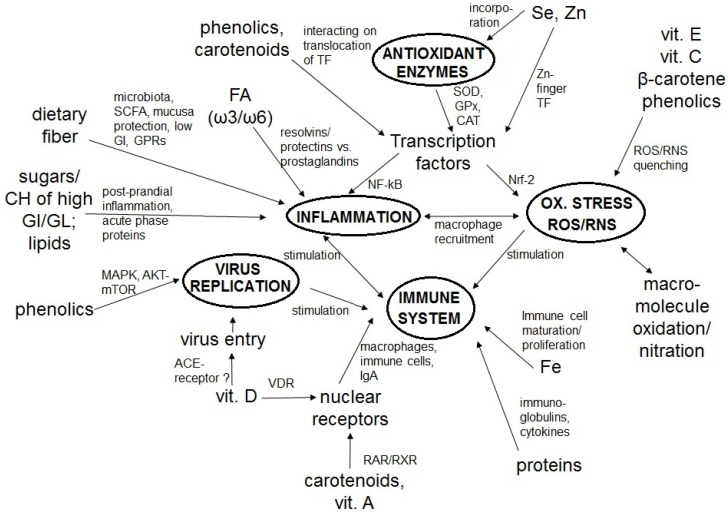
Schematic diagram showing interactions between selected dietary constituents, the immune system, and viral infection. *Abbreviations:* CH: carbohydrates; GALT: gut-associated lymphoid tissue; GPRs: G-protein-coupled receptors; FA: fatty acids; GI/GL: glycemic index/load; RAR/RXR: retinoic acid receptor/retinoid X receptor; SCFA: short-chain fatty acids; TF: transcription factors; VDR: vitamin D receptor.

**Table 1 nutrients-12-01562-t001:** Selected studies associating dietary constituents with viral or other infection risk and symptoms.

Constituent	Study Design	Description	Main Findings	Ref
Proteins	Human cross-sectional study	23 elderly patients subjected to influenza vaccination and measurement of their nutrient status.	Total protein status (determined by questionnaire) was slightly lower (*p* < 0.05) in influenza vaccine non-responders vs. responders (66 vs. 69 g/L). Similar results were found for iron, proposing that immune-response was compromised by poor nutrient status in this elderly population.	Fulop et al., 1999 [78]
	Animal study (mice)	Group receiving a diet adequate in protein (AP; 18% of energy) vs. group receiving very low protein (VLP; 2%) for 3 weeks. Both groups were subjected to influenza infection.	Higher mortality in the VLP group (*p* < 0.001) vs. AP, 25 d post infection (p.i.). The AP vs. the VLP group showed a decreased virus titer by day 9 (*p* < 0.001) and an efficient clearance within 12 d (*p* < 0.001). %age of NK cells in lungs were reduced (*p* < 0.01) in VLP vs. AP group with higher (*p* < 0.001) neutrophil proportions in response to infection with influenza virus in each group, respectively. The VLP group had less influenza NP-specific CD8^+^ T cells at days 8 (*p* < 0.05), 15 (*p* < 0.05), and 30 (*p* = 0.001). Switching VLP to AP diet improved CD4^+^ and CD8^+^ T cell subset levels on days 8 (*p* < 0.01), 15 (*p* < 0.01), and 30 (*p* < 0.01) and increased IFN-γ (*p* < 0.001).	Taylor et al., 2013 [79]
	Animal study (mice)	Mycobacteria-infected mice fed 2% protein diet vs. control group receiving 20% protein diet for up to 30 days.	100% of malnourished mice (fed 2% protein diet) succumbed to *M. tuberculosis* infection within 66 d p.i. Malnourished mice had a reduced expression of IFN-γ, TNF-α, and iNOS in the lungs. A mortal infection of *M. tuberculosis* in malnourished animals was reversed upon re-feeding with the 20% protein diet.	Chan et al., 1996 [80]
Lipids	Animal study (mice)	Mice infected with H5N1 virus treated with omega-3 polyunsaturated fatty acid-derived lipid mediator protectin D1 (PD1), given 3 times i.v.	H5N1 virus pathogenicity decreased with higher levels of PD1. PD1 inhibited virus replication (*p* < 0.001) via influenza virus nucleoprotein mRNA expression at day 2 p.i. PD1 treatment within 12 h improved the survival (*p* < 0.05) and pathology of severe influenza (*p* < 0.001).	Morita et al., 2013 [81]
	Animal study (mice)	Influenza A virus (IAV) infected mice fed with a high-fat (HF, 40% of energy) vs. low-fat (LF, 12% of energy) diet for 10 weeks.	HF mice were more susceptible to respiratory disease after IAV infection than were LF mice, with lower blood oxygen saturation (*p* < 0.05) and an increase in pulmonary viral load (*p* < 0.05). Decreased pro-inflammatory response to IAV in the serum of HF mice vs. LF for IL-6, IFN-γ, IFN-α, and IP-10 (*p* < 0.05). Antiviral response in the heart was reduced in HF mice after IAV infection, where higher viral loads were detected in the hearts of HF vs. LF mice (*p* < 0.05). Correlation between IAV-infected HF mice and viral infection in the heart, left ventricular mass, and thickening of the left ventricular wall, characterized by increased HIF-1α compared to LF group (*p* < 0.05).	Siegers et al., 2020 [82]
	Animal study (mice)	High-fat diet (HFD) 60% or regular-fat diet (RFD) 5% fat, administered to 4-week old mice for 10 weeks. Influenza vaccination was conducted after 10 weeks.	Functionality of macrophages was diminished after diet-induced obesity (*p* < 0.001) via lower CD86-expressing macrophages, lower release of IL-6 and TNF-α, increased Th1 cell subpopulation, and reduced proportion of Treg cells. Vaccination-induced antibody production was decreased in animals receiving HFD vs. RFD (*p* < 0.001)	Cho et al., 2016 [83]
Lipids, carbo-hydrates	Animal study (mice)	Feeding mice with ketogenic, i.e., low carbohydrate diet (KD, 90% fat) vs. standard high-fat (60% fats, 20% lipids) diet (HFD) for 7 d before influenza A virus (IAV) infection.	KD protected mice from lethal IAV infection and disease (*p* < 0.05) compared to HFD-fed mice. KD resulted in an expansion of T-cells (*p* < 0.001), compared with the HFD group. KD-fed mice had better blood O_2_ saturation (*p* < 0.001). KD diet was significantly related to improved antiviral resistance (*p* < 0.001).	Goldberg et al., 2019 [84]
Fiber	Prospective human cohort study	Study evaluating dietary fiber intake versus health outcomes. n = 219,123 men and 168,999 women, aged 50–71 y. 9 y follow-up.	Consumption of dietary fiber correlated with lowered mortality from infectious and respiratory diseases. Per 10 g/d increase in dietary fiber, the multivariate RRs for infectious and respiratory diseases were 0.66 (CI: 0.52–0.84) and 0.82 (CI: 0.74–0.93) in men and 0.61 (CI: 0.44–0.85) and 0.66 (CI: 0.56–0.78) in women, respectively.	Park et al., 2011 [85]
	Animal study (mice)	High-fiber diet (HFD)-fed mice vs. control group, subjected to viral influenza infection	Intake of dietary fiber improved influenza by prolonged survival (*p* < 0.05) and ameliorated clinical scores (*p* < 0.001). After 7 d, HFD-fed mice with high-dose infection had better lung function as shown by reduced pulmonary resistance (*p* > 0.01) and enhanced compliance in response to methacholine (*p* > 0.01). In HFD-fed mice, the excessive neutrophil influx into the airways was inhibited by blunted levels of CXCL1, produced by lung monocytes and macrophages (*p* < 0.001) vs. controls. Increased antiviral immunity by dietary fiber through CD8^+^ T cell activation (*p* < 0.01) vs. controls. (HFD)-fed mice showed enhanced adaptive immunity by changed CD8^+^ T cell metabolism (*p* < 0.05).	Trompette et al., 2018 [86]
	Animal study (mice)	Fiber-free diet group (LFD) vs. control group for up to 40 d, subjected to infection with mucosal pathogen *Citrobacter rodentium*	Low fiber intake resulting in increases in mucus-degrading microbiota and enhanced lethal colitis cases (*p* < 0.05).	Desai et al., 2017 [87]
Vitamin A	Meta-analysis of RCTs.	Effects of vitamin A supplementation on acute lower respiratory tract infections (LRTI). 10 studies (n = 33,179 children).	Though some individual studies demonstrated a positive effect of vitamin A supplementation on LRTI, in pooled analyses, there was no effect of vitamin A supplementation on acute LRTI incidence or prevalence of symptoms.	Chen et al., 2008 [88]
	Meta-analysis of RCTS.	Assessment of vitamin A supplementation on acute respiratory infection. 5 studies (n = 2177 children (1067 children under intervention, 1110 control).	Faster recovery from infection symptoms due to vitamin A, no differences in the placebo group: fever: OR: 0.03, CI: −0.10–0.17; oxygen requirement: OR: −0.08, CI: −0.31–0.16; increased respiratory rate: OR: −0.09, CI: −0.38 –0.19; hospital stay duration: OR: −0.06, CI: −0.52–0.40.	Brown and Roberts 2004 [89]
Vitamin D	Retrospective human study	Study determining mortality patterns of COVID-19 and associated factors: Special focus on vitamin D status. 2 cohorts of 780 cases with confirmed infection of SARS-CoV-2 in Indonesia.	Vitamin D status is strongly associated with COVID-19 mortality (adjusted for age, sex, and comorbidity) (*p* < 0.001). Individuals with insufficient vitamin D status were ca. 12.6 as likely to die (OR 12.55).	Reharusun et al., 2020 [90]
	Meta-analysis of RCTs	Assessment of vitamin D supplementation on respiratory tract infections. 5 clinical trials (n = 964 participants).	Significantly fewer respiratory tract infections were observed following a vitamin D supplementation. (OR: 0.58, CI: 0.417–0.812). In clinical trials there were beneficial effects on events of infections due to vitamin D supplementation in children (OR: 0.58, CI: 0.416–0.805) and adults (OR: 0.65, CI: 0.472–0.904).	Charan et al., 2012 [91]
	Meta-analysis of RCTs	Assessment of vitamin D supplementation on respiratory tract infection (RTI). 11 placebo-controlled studies (RTCs) (n = 5660 patients).	Vitamin D had protective effects against RTI (OR: 0.64; CI, 0.49- 0.84). This was more pronounced by individual daily dosing compared to bolus doses (OR = 0.51 vs. OR = 0.86, *p* = 0.01).	Bergman et al., 2013 [92]
Vitamin E	Humans, RCT	Assessment of vitamin E supplementation and community acquired pneumonia. n = 7469 men 50–69 y.	Lower incidence of pneumonia in individuals receiving vitamin E supplements (RR: 0.28; CI: 0.11–0.69).	Hemila, 2016 [93]
Vitamin C	Meta-analysis of RCTs	Supplementation trials with vitamin C and observation of cold symptoms. 9 randomized controlled trials (n = 5500) in children (3 months–18 y of age).	Daily supplementation in vitamin C with extra doses reduced the time of having a common cold (mean difference = −0.56, 95% confidence interval (CI) (−1.03, −0.10)), fever (mean difference = −0.45, 95% CI (−0.78, −0.11)) and chest pain (mean difference = −0.40, 95% CI (−0.77, −0.03)).	Ran et al., 2018 [94]
B-vitamins	Human cross-sectional study	Observation of inflammation markers and nutrient status. HIV infected participants (n = 180 men, 134 women; 18–60 y).	Serum CRP concentrations were inversely associated with increased vitamin B intake including niacin, pyridoxine, and cobalamin (*p* for trend *p* < 0.01, *p* < 0.05 and *p* = 0.037, respectively) in men. Trends were observed in women.	Poudel-Tandukar et al., 2016 [95]
Zinc	Human double-blinded RCT	Patients in the zinc group (n = 50) received lozenges (13.3 mg of zinc gluconate) as long as they showed cold symptoms. Patients in the placebo group (n = 50) received 5% calcium lactate pentahydrate.	A faster decrease of the cold symptoms (median, 4.4 d vs. 7.6 d; *p* < 0.001), e.g., fewer days with coughing (median, 2 d compared with 4.5 d; *p* < 0.05), hoarseness (2 and 3 d; *p* < 0.05), headache (2 and 3 d; *p* < 0.05), nasal congestion (4 and 6 d; *p* < 0.01), and sore throat (1 and 3 d; *p* < 0.001) were found in the intervention group, supplemented with zinc, in comparison with the placebo group.	Mossad et al., 1996 [96]
Iron	Animal trial (Wistar rats)	Administration of low iron diet (4–5 mg powder), medium iron diet (15 mg), control group (35 mg) and normal iron intake diet group. At week 4, rats received injection of inactivated porcine influenza vaccine (HswIN1).	Following immunization, anemic rats exhibited decreased (*p* < 0.05) antibody titer vs. controls. Antibody synthesis was preserved in moderate iron deficiency, but was hampered by severe anemia.	Dhur et al., 1990 [97]
Selenium	Human randomized, double-blinded RCT	Evaluation of response to influenza vaccine. 12-weeks follow up. n = 119 (50–64y) 6 intervention groups: 50, 100, or 200 mgSe/day, meals with Se-enriched onions (50 mg se/day), unenriched onions and placebo.	SEPS1 mRNA (marker of inflammation) increased (*p* < 0.05) after one week of vaccine administration, being dependent on the dose of Se per each intervention arm.	Goldson 2011 [98]
Polyphenols	Animal study (mice)	Evaluation of effect of polyphenol extract from *Cistus Incanus* on avian influenza Aviurs (H7N7). Inbred female Balb/c and C57Bl/6 mice at the age of 6–8 weeks.	The polyphenol extract helped mice to not contract avian influenza, and to not alter bronchiole epithelial cells, as well as to keep constant the body temperature and the gross motor activity.	Droebner et al., 2007 [99]
Carotenoids	Longitudinal study with infants	Observation of β-carotene in plasma. 194 HIV-infected infants.	β-Carotene was related to increased risk of death during HIV infection (OR: 3.16, CI: 1.38 to 7.21; *p* < 0.01).	Melikian et al., 2001 [100]

Abbreviations: AP: adequate protein; Balb/c: albino mouse strain; CD-86: cluster of differentiation 86; CRP: C-reactive protein; CXCL1: The chemokine (C-X-C motif) ligand 1; H5N1- influenza A virus subtype H5N1; H7N7: influenza A virus subtype H7N7; HF: high fat; HFD: high-fat diet; HFD: high-fiber diet; HswIN1: swine influenza virus; IAV: influenza A virus; IFN-α/γ: interferon α/γ; IL6: interleukin 6; LF: low-fat; LRTI: lower respiratory tract infections; iNOS: inducible nitric oxide synthase; KD: ketogenic diet; NK: natural killer cells; P1- protectin D1; RCT: randomized controlled trial; RFD: regular-fat diet; RTI: respiratory tract infection; SEPS1: selenoprotein S; TNF-α: Tumor necrosis factor alpha; VLP: very-low protein.

**Table 2 nutrients-12-01562-t002:** Examples of dietary sources.

Constituent	Major Food Sources	Quantity
Protein (g/100 g or mL)	Meat products:	
	Beef	25.3
	Chicken	19.3
	Egg white	11
	Dairy products:	
	Yogurt	3.5
	Milk	3.1
	Cereals, roots, and tubers:	
	Potatoes	2.4
	Quinoa	4.4
	Legumes:	
	Soybeans (raw)	25.9
Lipids (mg/100 g)	Fruits and vegetables:	
(High in omega-3)	Chia seeds	1783
	Edamame	361
	Avocado	111
	Animal source:	
	Salmon	2314
	Tuna Fish	1337
	Whole grain food:	18
	Oatmeal	
Carbohydrates (g/100 g)	Fruits and vegetables:	
	Blueberries	14.5
	Figs	19.2
	Summer squash	3.8
	Whole grain food:	
	Oatmeal	12
	Whole-wheat bread	42.7
	Legumes:	
	Black beans	23.7
Fiber (g/100 g)	Fruits and vegetables:	
	Chia seeds	34.4
	Soybeans	1.1
	Orange	2.4
	Brussel Sprouts	3.8
	Legumes:	
	Lentils	7.9
	Chickpeas	7.6
Vitamin A (µg/100 g)	Fruits and vegetables:	
	Carrots (raw)	835
	Cantaloupe	169
	Mango	54
	Animal source:	
	Salmon	13
	Eggs	160
Vitamin D (µg/100 g)	Vegetables:	
	Portabella mushrooms	0.33
	Animal source:	
	Salmon	14.4
	Chicken	0.14
	Egg (whole, raw)	2.1
	Low fat yogurt	0.03
Vitamin E (mg/100 g)	Fruits and vegetables:	
	Sunflower seeds	35.2
	Nuts, almonds	25.6
	Blueberries	0.6
	Kiwi	1.5
	Broccoli	0.8
Vitamin C (mg/100 g)	Fruits and vegetables:	
	Oranges	53.2
	Broccoli	89.2
	Brussel sprouts	85
	Lemon	53
	Cauliflower	48.2
Vitamins B_6_ (mg/100 g)	Plant source:	
	Peanuts	0.5
	Lentils	0.2
	Animal source:	1
	Tuna fish	0.4
	Mollusks (raw)	
Vitamin B_12_ (µg/100 g)	Animal source:	
	Mollusks (raw)	14.1
	Plain yogurt	0.4
	Chicken breast	0.2
Zinc (mg/100 g)	Plant source:	
	Pumpkin and squash seeds	7
	Nuts	3.1
	Soybeans	1.2
	Animal source:	
	Beef	7.4
	Mollusks (raw)	16.6
	Lamb	4.9
Iron (mg/100 g)	Fruits and vegetables:	
	Apricots (dehydrated)	2.7
	Tomatoes (cherry)	0.3
	Peas	1.5
	Sunflower seeds	5.3
	Animal Source:	
	Mollusks	5.1
	Egg	1.8
	Veal (ground)	1.4
Copper (mg/100 g)	Vegetables:	
	Cashew nuts	2.2
	Tofu	0.4
	Mushrooms	0.3
	Animal Source:	
	Beef	0.2
	Oyster	1.6
	Cereals, roots, and tubers:	
	Sweet potato	0.3
	Quinoa	0.2
Selenium (µg/100 g)	Plant source:	
	Sunflower seeds	53
	Coconut meat	17
	Animal Source:	
	Mollusks	77
	Salmon	47
	Turkey, ham	37
Polyphenols	Flavanone	
(mg/100 g)	Oranges (raw)	42.6
	Grapefruit juice	31.2
	Anthocyanidin	
	Blueberries (raw)	163.5
	Strawberries (raw)	33.6
	Flavan-3-ol	
	Black tea	115.3
	Apple juice	6
Carotenoids (mg/100 g)		α-Carotene
	Mixed frozen vegetables	1.4
	Tomatoes	0.08
	Tangerines	0.08
		β-Carotene
	Spinach	10.8
	Kale	9
	Cantaloupe	3

Source: United States Department of Agriculture.

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
