# Peer review of "Strengthening the Immune System and Reducing Inflammation and Oxidative Stress through Diet and Nutrition: Considerations during the COVID-19 Crisis"

_nutrients, 2020, doi:10.3390/nu12061562_

Round 1

Reviewer 1 Report

In this paper, the authors have emphasized the importance of the optimal nutrient status, in order to reduce inflammation and oxidant stress effectively.

The authors have described the method of the immune system for overcoming the COVID-19 crisis from many bibliographic considerations.

Moreover, the role of the food factors to an immune system have been described in detail.

I think that this review can provide with the useful information to all the medical staffs.

Author Response

General remark by the authors:

We extend our gratitude to the editorial staff and the reviewers for their careful reading and comments. In addition to the responses below, some further extra corrections have been introduced in the manuscript, such as expanding the paragraph on vitamin D and dietary fiber slightly, updating references, and improving on minor errors present in the manuscript.

Reviewer 1 :

In this paper, the authors have emphasized the importance of the optimal nutrient status, in order to reduce inflammation and oxidant stress effectively. The authors have described the method of the immune system for overcoming the COVID-19 crisis from many bibliographic considerations. Moreover, the role of the food factors to an immune system have been described in detail. I think that this review can provide with the useful information to all the medical staffs.

Reply: we appreciate the reading of the article by the reviewer and the kind comments.

Reviewer 2 Report

Overall, this was a comprehensive review about macro/micronutrients and their role to reduce inflammation/oxidative stress. It was difficult to see the strong connection with the gut, aside from when it was discussed in the fiber part, so unsure if this is necessary with limited discussion. A few other minor comments to consider:

Abstract: As this was a review, discuss if all macro/micronutrients were viewed or the ones of focus. Please include the main results – vitamin C at 1000mg/day reduces infection complications, etc.

Section 1 and 2

This is a comprehensive overview of inflammation and oxidative stress, but may suggest to have the description about how COVID-19 or other viral infections contribute to inflammation/oxidative stress first and then about diet.

Table 1 varies in the description. For example, in the first human study, the description just says the population and influenza vaccination, but what specifically was this study about? Similar to the descriptions included the animal study please specify it for those studies. The discussion about lipids was not brought up in the first section.  

The Goldberg 2019 study mentioned high fat/high carbohydrate diet. Based on the results presented, this appeared to be more on the focus of lipids as opposed to carbohydrates as it is not clear if the carbohydrates were simple or complex.

3.1 Protein

Please clarify what is meant by low-protein intake. Even though soy and other plant-based proteins are lower in biological value than meats, overall, they have been shown to reduce inflammation compared to lean meats.  

3.1.3 Carbs and fiber

Mention what percentage lower dietary fiber intake is compared to the recommendations. At least for table 1, the findings focused more on inflammation/oxidative stress as opposed to gut microbiota and that role. Therefore, suggest to include more of those studies in that table.

Recommend a table the summarizes the amount that appears to be most effective in reducing inflammation/oxidative stress.

Table 2 needs to be clarified. It mentions the unit next to the macro/micronutrient, then in quantity/100 g or ml. Therefore, in 100g of beef there is 25.3g of protein. However, to help the reader, it may be beneficial to have a macronutrient table and then a micronutrient table or at least display it differently.

Author Response

Kind regards,

Torsten Bohn

Reviewer 3 Report

The immune system plays a vital role in controlling inflammatory process which could be the root cause for many diseases.Infection such as those observed in recent times by COVID-19 pandemic is creating untold sufferings all over the world.Currently no vaccines or treatment are available for tackling COVID-19 infection. However strengthening innate immune system could offer some help in fighting this viral infection. Many dietary components could play a significant role in boosting immunity,reducing inflammation and other pro inflammatory markers such as Reactive Oxygen Species which may help in tackling the spread of COVID-19 infection.In the present review the authors have recorded the impact of macro- and micro nutrients from dietary sources which can be promoted to boost immunity to tackle viral infection. An account of data from meta analysis and also the molecular mechanisms by which the dietary components are showing promise in reducing inflammatory onslaught is also high lighted in this review. The subject matter of this review is timely.

Author Response

General remark by the authors:

We extend our gratitude to the editorial staff and the reviewers for their careful reading and comments. In addition to the responses below, some further extra corrections have been introduced in the manuscript, such as expanding the paragraph on vitamin D and dietary fiber slightly, updating references, and improving on minor errors present in the manuscript.

Reviewer 3:

The immune system plays a vital role in controlling inflammatory process which could be the root cause for many diseases. Infection such as those observed in recent times by COVID-19 pandemic is creating untold sufferings all over the world. Currently no vaccines or treatment are available for tackling COVID-19 infection. However strengthening innate immune system could offer some help in fighting this viral infection. Many dietary components could play a significant role in boosting immunity, reducing inflammation and other pro inflammatory markers such as Reactive Oxygen Species which may help in tackling the spread of COVID-19 infection. In the present review the authors have recorded the impact of macro- and micro nutrients from dietary sources which can be promoted to boost immunity to tackle viral infection. An account of data from meta analysis and also the molecular mechanisms by which the dietary components are showing promise in reducing inflammatory onslaught is also highlighted in this review. The subject matter of this review is timely.

Reply: We thank the reader for reading our manuscript and appreciate his/her evaluation.